# Characterization of Sand and Dust Pollution Degradation Based on Sensitive Structure of Microelectromechanical System Flow Sensor [note 1]

**DOI:** 10.3390/mi15050574

**Published:** 2024-04-26

**Authors:** Jinchuan Chen, Xiao Wen, Qinwen Huang, Wanchun Ren, Ruiwen Liu, Chunhua He

**Affiliations:** 1School of Information Engineering, Southwest University of Science and Technology (SWUST), 59 Qinglong Road, Mianyang 621010, China; swustcjc@mails.swust.edu.cn (J.C.); wenxiao@mails.swust.edu.cn (X.W.); 2Science and Technology on Reliability Physics and Application Technology of Electronic Component Laboratory, China Electronic Product Reliability, and Environmental Testing Research Institute, Guangzhou 510610, China; 3Institute of Microelectronics Chinese Academy of Sciences, Beijing 100017, China; liuruiwen@ime.ac.cn; 4School of Computer, Guangdong University of Technology, Guangzhou 510006, China; hechunhua@pku.edu.cn

**Keywords:** microelectromechanical systems, failure mechanisms, predisposing factors, electrostatic effects

## Abstract

The effect of sand and dust pollution on the sensitive structures of flow sensors in microelectromechanical systems (MEMS) is a hot issue in current MEMS reliability research. However, previous studies on sand and dust contamination have only searched for sensor accuracy degradation due to heat conduction in sand and dust cover and have yet to search for other failure-inducing factors. This paper aims to discover the other inducing factors for the accuracy failure of MEMS flow sensors under sand and dust pollution by using a combined model simulation and sample test method. The accuracy of a flow sensor is mainly reflected by the size of its thermistor, so in this study, the output value of the thermistor value was chosen as an electrical characterization parameter to verify the change in the sensor’s accuracy side by side. The results show that after excluding the influence of heat conduction, when sand particles fall on the device, the mutual friction between the sand particles will produce an electrostatic current; through the principle of electrostatic dissipation into the thermistor, the principle of measurement leads to the resistance value becoming smaller, and when the sand dust is stationary for some time, the resistance value returns to the expected level. This finding provides theoretical guidance for finding failure-inducing factors in MEMS failure modes.

## 1. Introduction

In the field of microelectromechanical system (MEMS) flow sensors [1], gas sensors [2], and optical microelectromechanical systems (MEMS), the sensitive structures [3] or components of the devices need to be directly or indirectly exposed to the working environment, thus becoming susceptible to functional contamination in the environment by dust contaminants [4]. This contamination can lead to the degradation of device performance, affecting its operational life and stability. In addition, these contaminants may also lead to the degradation of the electrical characteristic parameters of the product during operation, thus posing a significant challenge to the reliability of MEMS devices. Existing dust testing standards, such as IEC 60529-2013, GB/T 4208-2017, GB/T 2423.37-2006, and GJB 150.12A-2009, are not applicable to the testing of MEMS sensitive structures for dust contamination testing [5,6,7,8]. In order to investigate the specific effects of dust particles on the failure modes of MEMS devices [9] and to assess their reliability, we propose a dust particle contamination experiment. This experiment aims to explore the effect of electrostatic forces [10] generated by friction between dust particles on the electrical characteristics of MEMS sensors [11], and to provide a guiding principle for investigating the fundamentals of MEMS failure. In previous studies, Yuan Changrong [12] specifically analyzed the extent to which the presence or absence of contaminants affects the flow velocity on the surface of the sensor chip through fluent fluid simulation software, and Wei Yujin [13] conducted modeling and simulation experiments on contamination in the sensitive region of a MEMS flowmeter using COMSOL software 6.0. The analysis focused on the effect of different thicknesses of dust on the sensitive region. Bai Xuejie [14] quantitatively analyzed the effects of particle size, collision velocity, particle type, and contact mode on the potential and morphological changes in silica surfaces. However, none of these studies explored the internal inducing factors of their pollution failure mechanisms in detail. The most studied aspect was the effect of dust cover on heat transfer [15], as shown in Figure 1, while the device reliability guidelines needed to be more comprehensive. Therefore, in this paper, we explore in detail the effects of dust particle size and relative humidity on the electrical characteristics of MEMS devices in sand and dust contamination experiments. In the second part, we delve into the principles of dust settling, electrostatic dissipation, and multimeter measurements. In the third part, we comprehensively elucidate the mechanisms of the effects of dust particle size and relative humidity on the electrical characteristics of devices through theoretical derivation and descriptive simulation. In the fourth part, we derive the output data through relevant pollution tests and fit them in comparison with the simulation data. Finally, in Part V, we present the research results with in-depth discussions, which are compared and analyzed with the existing research results.

## 2. Design and Principle

### 2.1. Dust-Falling Principle

When a MEMS thermistor [16] is placed in a contaminated test chamber, dust pollutants gradually accumulate on the sensitive structure. As time goes on, the deposition amount increases, leading to continuous changes in the electrical characteristics of the sensor. In order to evaluate the influence of different particle sizes and relative humidity on the device’s electrical characteristics, this study utilized MATLAB software R2022b to establish a simulation model. By incorporating a particle deposition model, we were able to simulate the impact of frictional electrostatics caused by dust pollutants on the electrical characteristics of the sensitive structure, and combine the simulation data to perform reliability modeling and assessment.

The dimensionless deposition rate of pollutants was calculated using the particle deposition model proposed by Fan F G et al. [17], as shown in Equation (1):(1)$$u_{d}^{+}={\left\{\begin{array}{l} 0.084Sc^{-2\backslash 3}+\frac{1}{2}\left[\frac{{\left(0.64k^{+}+\frac{d^{+}}{2}\right)}^{2}+\frac{\tau^{+2}g^{+}L_{1}^{+}}{0.01085\left(1+\tau^{+2}L_{1}^{+}\right)}}{3.42+\frac{\tau^{+2}g^{+}L_{1}^{+}}{0.01085\left(1+\tau^{+2}L_{1}^{+}\right)}}\right]\\ \;\;\;\;\;\times \left[1+8e^{-{\left(\tau^{+}-10\right)}^{2}/32}\right]\frac{0.037}{1-\tau^{+2}L_{1}^{+}\left(1+\frac{g^{+}}{0.037}\right)},u_{d}^{+}<0.14\\ 0.14\;\;\;\;\;\;\;\;\;\;\;\;\;\;\;\;\;,\mathrm{Other}\;\mathrm{situations} \end{array}\right.}^{1/\left(1+\tau^{+2}L_{1}^{+}\right)}$$

His model belongs to the category of empirical models, which have been developed by referencing experimental data, thereby ensuring a remarkable alignment with real-life sedimentation processes. The model encompasses a multitude of parameters, including dimensionless sedimentation velocity, Reynolds number, and Schmidt number. The dimensionless sedimentation velocity serves as a crucial indicator of turbulent friction velocity, frequently employed to evaluate the magnitude of turbulence. The Reynolds number, a dimensionless quantity, serves as a discriminant for the flow state of viscous fluids [18]. Conversely, the Schmidt number, another widely utilized dimensionless parameter, plays a pivotal role in quantifying the extent of molecular diffusion, exerting a profound influence on the diffusion zone.

### 2.2. Principle of Frictional Electrostatic Dissipation

Firstly, let us assume that under the same humidity conditions, the dissipation coefficient [19] of like charges remains constant. The dissipation equation in a two-dimensional scale [20] can be expressed as:(2)$$\frac{\partial U}{\partial t}=D(\frac{\partial^{2}U}{\partial x^{2}}+\frac{\partial^{2}U}{\partial y^{2}})$$

In this context, U represents the indicative potential of the sample, while t signifies the dissipation time, and D represents the dissipation coefficient. The simple numerical solution within the simulated range is obtained through the discretization function $u_{i,j}^{\left(n\right)}$, which corresponds to the continuous equation U(x,y,t) with the inclusion of 0≤x≤20 μm, 0≤y≤20 μm, x=iΔy, t=nΔt. By employing finite difference approximation, the desired results can be achieved:(3)$$\frac{u_{i,j}^{\left(n+1\right)}-u_{i,j}^{\left(n\right)}}{\Delta t}=D\left[\frac{u_{i,j}^{\left(n+1\right)}-2u_{i,j}^{\left(n\right)}+u_{i-1,j}^{\left(n\right)}}{{\left(\Delta x\right)}^{2}}+\frac{u_{i,j}^{\left(n+1\right)}-2u_{i,j}^{\left(n\right)}+u_{i-1,j}^{\left(n\right)}}{{\left(\Delta y\right)}^{2}}\right]$$

In this context, $u_{i,j}^{\left(n+1\right)}$ and $u_{i,j}^{\left(n\right)}$, respectively, represent the surface potential at a point with a step size of n+1 and n. Δx and Δy represent the positional changes of the charge in the lateral and longitudinal directions during the two-dimensional dissipation process. By using the surface potential at a step size of n and the predetermined dissipation coefficient of D, the surface potential at a step size of n+1 can be calculated:(4)$$u_{i,j}^{\left(n+1\right)}=u_{i,j}^{\left(n\right)}+D\Delta t\left[\frac{u_{i,j}^{\left(n\right)}-2u_{i,j}^{\left(n\right)}+u_{i-1,j}^{\left(n\right)}}{\Delta x^{2}}+\frac{u_{i,j}^{\left(n\right)}-2u_{i,j}^{\left(n\right)}+u_{i-1,j}^{\left(n\right)}}{\Delta y^{2}}\right]$$

The initial value of the surface potential of the sample is simulated based on the surface potential distribution data at 0 min of dissipative time. In the simulation process, the boundary conditions are set to maintain a constant surface potential value at each boundary point, consistent with the experimentally measured values. To ensure the stability and accuracy of the simulation, it is necessary to ensure that the maximum time interval between each time step in the simulation is smaller than a certain threshold:(5)$$\Delta t=\frac{1}{2D}\frac{\left(\Delta x\Delta y\right)}{{\left(\Delta x\right)}^{2}+{\left(\Delta y\right)}^{2}}$$

### 2.3. Principle of Measuring Electrical Characteristics

In this experiment, we opted for resistance value output as the electrical characteristic parameter to evaluate the properties of the tested sample. To accomplish this objective, we required the use of a precise multimeter [21] for resistance measurement. The principle of resistance measurement using a multimeter is based on Ohm’s law, in which the internal voltage source of the multimeter is derived from an internal battery. The resistance value of the multimeter comprises several crucial components, including the tested resistance, adjustable resistance (for different range settings of internal resistance), fixed resistance, and zero-adjustment resistance. By measuring the current, we can accurately calculate the resistance value. Specifically, we can employ the following formula for calculation: I = U/(Rg + Rfixed + Rzero + Rtest), where U represents the voltage of the internal battery, Rg is the resistance of the meter head, Rfixed is the fixed resistance connected in series with the meter head, Rzero is the adjustable zero-adjustment resistance, and Rtest is the resistance being measured. To enhance comprehension of the measurement principle, a measurement schematic is provided in Figure 2.

The current reaches its maximum value when the resistance Rx in the circuit is zero. By adjusting the resistance R, the angle of deflection of the pointer on the measuring instrument can be made to reach the full-scale value, indicating a current value of I0 = E/R in the circuit. As the resistance Rx being measured increases, the current I = E/(R + Rx) decreases gradually, decreasing the deflection angle of the pointer. Consequently, the resistance value scale on the multimeter dial is marked in the opposite direction. The presence of dust pollution can be observed in Figure 3:

When the sand particles descend and come into contact with the thermistor, the sand particles possess initial kinetic energy, causing mutual friction and generating a minute electric current that enters the thermistor. When measuring the resistance using a multimeter, the measured current value in the circuit will be higher than the actual current value:(6)$$I_{A}=I_{F}+I_{Z}$$
(7)$$I_{Z}=\frac{E}{R+R_{x}}$$

In the provided equation, the variable $I_{A}$ represents the resultant current obtained from the measurement, while the variable $I_{F}$ signifies the static electricity current produced by the friction between individual sand particles. On the other hand, the variable $I_{Z}$ denotes the genuine current traversing through the resistor under examination. It is important to note that the magnitude of the final measured current surpasses the actual current flowing through the resistor. Following the principles of multimeter measurement elucidated above, this leads to a lower resistance value being recorded compared to its actual value. However, when the sand particles remain undisturbed for a particular duration, the static electricity current gradually diminishes, ultimately approaching zero. Consequently, the resistance measurement obtained during this state reflects the genuine resistance value.

## 3. Emulation

### 3.1. Principle of Measuring Electrical Characteristics

Based on the expression of the parameters in the particle deposition model [22,23] given by Equation (1), we developed a set of MATLAB programs to visualize the simulation of the deposition process of sand dust particles on a resistor. This program requires several input values, including the actual atmospheric pressure, ambient temperature, roughness of the deposition surface, diameter of the particle molecules, particle density, and acceleration, among others. With these input values, we can simulate the deposition of particles on the resistor surface more accurately.

According to the provided information, the schematic diagram of the final program simulation results is shown in Figure 4a,b. In Figure 4a, we can observe the initial step of the particle deposition process. In a normal distribution, sand particles are present in the space above the resistor in a normal distribution. The sizes of these particles follow a normal distribution ranging from 1 to 100 μm. Under the influence of wind speed and gravity, the particles start to gradually settle down. This process can be seen as the interaction between particle motion and forces in the air.

Figure 4b displays the situation after the completion of particle deposition. We can see that the sand particles have adhered to the surface of the resistor. At the same time, these sand particles still possess some initial kinetic energy during the settling stage. At this stage, friction occurs between the sand particles, leading to static electricity effects. This static electricity effect may cause attraction or repulsion between the particles, further affecting the distribution and adhesion of the particles.

### 3.2. Sand and Dust Pollution Simulation

This simulation utilized MEMS thermal resistors [24] as the testing units. The principle of the test is that under controlled environmental conditions, when the surface of the structure is covered by dust particles, tiny friction is generated between the particles, resulting in static electricity. This causes the ammeter in the measurement circuit to read a higher current value than the actual value, leading to a change in the measured resistance of the thermal resistor. In order to observe the impact of pollutants on the device’s electrical characteristics, we conducted a simulation and continuous data collection using MATLAB software, simulating the accumulation of dust. This allowed us to observe the effects of pollutants on the device in a short period of time. To study the different effects of particle size and relative humidity on the electrical characteristics of the thermal resistor, we designed an orthogonal experiment. The orthogonal factors include particle size and relative humidity. Through this design, we can more accurately understand the impact of these two factors on the thermal resistor and draw scientifically reliable conclusions.

In the simulation software MATLAB, a single change in relative humidity was made by controlling the size of the sand particles to be normally distributed, resulting in the experimental results in Figure 5 showing the significant effect of humidity on static electricity. By scrutinizing the graph, we can clearly see the direct effect of humidity change on static electricity. As humidity increases, we can observe a significant decrease in static electricity, which indicates a clear inverse relationship between humidity and static electricity.

In the simulation software MATLAB, the relative humidity size was normally distributed so as to ensure that the relative humidity variable would not affect the electrostatic effect. With only a single change in the size of the sand particles, we can carefully observe Figure 6 to understand the different sizes of silica particles induced by the change in the electrostatic effect. It is clear that larger particles produce a stronger electrostatic effect, which leads to an enhanced current. In contrast, smaller particles produce only a weaker electrostatic effect, which is proportional to particle size. This difference in electrostatic effect based on particle size is obvious.

The intricate depiction of Figure 7a in the three-dimensional plane unveils the intimate relationship between electrostatic effects and particle size. Meticulous observation reveals a significant enhancement of the electrostatic effect with a gradual increase in particle dimensions.

Based on the aforementioned analysis, we can conclude that the electrostatic effect of silica particles during the sedimentation process is influenced by both humidity and particle size. This research finding will significantly contribute to a more comprehensive understanding and effective control of the electrostatic effect in MEMS thermal resistors.

For the experimental procedure, we employed a multimeter to precisely measure the resistance of the dust-contaminated resistor precisely. As illustrated in Figure 2, the resistor was connected to the multimeter for accurate measurement. Furthermore, we also recorded the current magnitude and resistance value in the circuit to obtain a more comprehensive dataset. During the simulation process, we assumed an initial resistance value of 300 ohms and compressed the simulation time to a concise 60 s. Moreover, we conducted both single-variable simulations, where the environmental relative humidity was set at 50%, and multi-variable simulations, where we varied the humidity levels. This meticulous approach allowed us to effectively simulate the impact of humidity on resistance performance in a realistic environment.

Ultimately, we employed MATLAB for the simulation process and obtained compelling results, as depicted in Figure 7b. These results provide intricate details regarding resistance performance and the magnitude of current in the circuit. By meticulously analyzing and interpreting these data, we can gain profound insights into the resistance behavior shown in a dust-contaminated environment.

As depicted in Figure 7a, during the initial stage of dust reduction, the sand particles possess initial kinetic energy and undergo frictional motion, generating static electricity. According to the principle of static electricity dissipation, as indicated by Equations (2)–(5), the static electric current flows into the multimeter and forms a detection circuit with the resistor. This leads to a slight increase in the measured current within the circuit. Consequently, the sand particles cease their frictional motion and the static electricity effect dissipates, causing the static electric current to approach zero. Based on the principles of multimeter measurements and Equations (6) and (7) presented above, the measured resistance value slightly decreases, followed by a return to the normal range of resistance values.

As illustrated in Figure 7c, with the increment of relative humidity, a reduction in the increment of current in the measurement circuit is observed during the initial phase of dustfall. This phenomenon is in accordance with the measurement principle discussed earlier, where a corresponding decrease in resistance alteration is anticipated. The simulation conducted in this study further substantiates the impact of particle size and environmental relative humidity on the manifestation of electrostatic effects within the experimental investigation of dust particle pollution.

## 4. Pollution Test

### 4.1. Samples

This design utilizes a MEMS sensitive structure with a four-ended solid support beam structure. The four-terminal solid beam structure is commonly used for thermal MEMS devices such as thermal infrared detectors, thermally conductive gas sensors, miniature Pirani vacuum gauges, and other MEMS devices. Figure 8 illustrates a plan view and wafer diagram of a MEMS thermistor four-terminal solid support structure with overall dimensions of 2600 µm × 2800 µm. The thermistor was prepared on the cross-section of a four-terminal fixed beam using a coating and etching process. The shaded area has the size of a 1080 µm by 1080 µm etched cavity containing the spiral structure of two prepared thermistors. Pad1-4 are the pins on the silicon wafer, connected to the PCB board by gold wire bonding. Rt1-4 are the resistors connected to the four-ended fixed beams, which are used to provide voltage to the two thermistors. This gives us the design’s primary structure and the components’ layout.

The thermistor has an excellent cross-sectional structure, as shown in Figure 9a. It consists of several layers of support beams composed of different materials assembled on a silicon substrate. Specifically, the structure of the support beams consists of 0.5 µm of low-pressure chemical vapor-deposited silicon dioxide (LP-TEOS), 0.4 µm of low-pressure chemical vapor-deposited silicon nitride (LP-SiN), and 0.5 µm of low-pressure chemical vapor-deposited silicon dioxide (LP-SiO). The thermistor, on the other hand, is prepared from 0.18 µm platinum (Pt) and covered with a layer of 0.8 µm plasma-enhanced chemical vapor-deposited silicon nitride (PE-SiN). This structural design not only provides a stable support function but also ensures the performance and stability of the thermistor. As shown in Figure 9b, the three-dimensional schematic of the thermistor, the ground liner, the support beam, and the spiral structure resistor can be seen more clearly.

The thermistor has a very fine cross-sectional structure, as shown in Figure 10. It consists of several layers of support beams composed of different materials assembled on a silicon substrate. Specifically, the structure of the support beams consists of 0.5 µm of low-pressure chemical vapor-deposited silicon dioxide (LP-TEOS), 0.4 µm of low-pressure chemical vapor-deposited silicon nitride (LP-SiN), and 0.5 µm of low-pressure chemical vapor-deposited silicon dioxide (LP-SiO). The thermistor, on the other hand, is prepared from 0.18 µm platinum (Pt) and covered with a layer of 0.8 µm plasma-enhanced chemical vapor-deposited silicon nitride (PE-SiN). This structural design not only provides a stable support function, but also ensures the performance and stability of the thermistor.

As shown in Figure 11, on the PCB, by connecting the Pad1-4 pins, we can read the resistance value of the thermistor. At room temperature, we can use a digital multimeter or other devices to connect Pad1, Pad2 or Pad3, and Pad4 to read the resistance value. According to the test results, we can know that the resistance value obtained when connecting Pad1 and Pad2 or connecting Pad3 and Pad4 is 270 Ω. This value can help us understand the characteristics and performance of the thermistor.

### 4.2. Experimental Methods

In this experiment, a sand and dust pollution test platform (CET-CC1000) was chosen as shown in Figure 12, which continuously collects dust during regular operation to observe the effect of pollutants on the device parameters in a short time. The sand and dust contamination test of MEMS thermistors relies on a dust-blowing test chamber. The dust-blowing speeds of the dust-blowing test chamber were set from 18 m/s to 29 m/s, and the wind speed of dust reduction was not more than 0.2 m/s. The diameter of the dust was less than 149 micrometers, ensuring that the settling speed of the dust was 6 g/m^2^/d. Because of the limitations of the test conditions, the blowing chamber could not be adjusted to the size of the sand and dust particles. Therefore, the experiment could only be conducted by fixing the size of sand and dust particles and qualitatively adjusting the size. We proceeded by fixing the size of the sand particles, qualitatively increasing the environmental humidity, and then verifying the electrostatic effect and how it was affected by the level of environmental humidity.

As shown in Figure 13, the sample was placed in a beaker, and the beaker was placed inside the dust-blowing test chamber and led outside the chamber; the thermistor was connected to a digital external meter, and the resistance value of the thermistor was used as the test result. At the same time, a humidifier was placed in the dust blower box to change the humidity level of the dust-contaminated environment. There were two test samples, each with eight thermistors, one with the humidifier on and one with the humidifier off.

The specific test procedure was as follows:

(1) Put the beaker into the dust-blowing test chamber, record the current ambient temperature, read the resistance value of the test resistance, and record the initial resistance value of the test resistance;

(2) Set the blowing dust test box condition setting cycle as 1 h, blowing dust as 5 min, and dust reduction as 5 min;

(3) After the test box has run for 10 min, close the test box, and when the temperature is the same as the temperature in step (1), record the resistance value of the test resistor and put it on record.

### 4.3. Data Analysis

#### 4.3.1. Experimental Data

Due to the individual differences of the thermistors, the initial resistance values of the eight samples were all 270 ± 30 Ω. The group with the humidifier not turned on measured 12 points after 2 h of contamination testing, leading to the test results shown in Figure 14a. The test results show that the resistance value of the MEMS thermistor varies significantly within the first five test points of the test, and there is no noticeable fluctuation in the resistance value from 2 to 9 h. After one hour of contamination, the resistance value of the test resistor of the MEMS thermistor drops by up to 2.5 Ω, and the average drops by 2.075 Ω. The resistance value rises after two hours of contamination, and gradually returns to the expected resistance value. These data show that the failure of the sand particles on the thermistor is affected by the electrostatic effect of the sand particles, which leads to a smaller resistance value. However, with time, the movement of the sand particles tends to stabilize, and the electrostatic effect disappears. The resistance value of the thermistor is restored to the average level, which is in line with the theoretical derivation as well as the simulation analysis in the previous section.

The MEMS thermistor with the humidifier on was measured at a total of 12 points after 2 h of contamination testing, and the results are shown in Figure 14b. After 1 h of pollution, the maximum value of resistance change is 1.3 Ω, which is 1.2 Ω less than the resistance fluctuation value of the group without the humidifier turned on. These data also show that the failure effect of the sand particles on the thermistor occurs through the electrostatic effect of the sand particles, which in turn leads to the resistance value becoming more minor, but because of qualitative changes in the ambient relative humidity, the electrostatic effect of the sand particles decreases. The trend of the change in its resistance value decreases, which is consistent with the theoretical deduction of the sand particles as well as the theoretical deduction of the ambient humidity and the trend of the change in the resistance value. This is in line with the previous theoretical derivation of the electrostatic effect and the theoretical derivation and simulation analysis of the effect of environmental humidity on the size of the electrostatic effect.

#### 4.3.2. Comparison of Data Fitting

Because the resistance value of the test resistor could not be sampled continuously, only fixed-point sampling resistance values could be obtained, so the test sampled the lowest resistance value and the simulation of the lowest resistance value obtained for error comparison, as shown in Figure 15, different coloured dots represent different resistances, horizontal coordinates represent actual measured values, vertical coordinates represent simulated values.

By comparing the error between the simulated data and the measured data, it was found that the average error was less than 0.25%.

## 5. Conclusions

The microelectromechanical system thermistor designed in this study exhibited resistance drift during sand and dust contamination tests, and the change in resistance value reflected the effect of sand and dust particles on the sensitive structure of the microelectromechanical system. It was verified through simulation that the failure mode under the influence of sand and dust particle contamination exhibited resistance drift due to the friction between sand and dust particles caused by the initial kinetic energy, which generated an electrostatic current that dissipated into the resistor. The electrostatic current affected the test circuits in the measurement loop, causing the measured resistance value to drift while the actual resistance value remained constant. In addition, the simulation model was simulated by controlling the relative humidity change from ten to ninety per cent with a gradient difference of ten per cent, as well as controlling the size of the sand and dust particles from 10 µm to 100 µm with a gradient difference of 10 µm. The final simulation results showed that the magnitude of the electrostatic current was directly proportional to the size of the particles and inversely proportional to the relative humidity. In the subsequent experimental validation, it was verified that the inducing factor for the drift of the thermistor resistance value was the electrostatic effect of the sand particles. In addition, this study qualitatively verified that an increase in ambient humidity attenuated the electrostatic effect; by increasing the ambient humidity, the initial average resistance value of 270 Ω over the test was reduced by 0.775 Ω. However, the effect of sand grain size on the electrostatic effect could not be verified because the dust blower was unable to control the sand grain size. This study provides a basis for identifying and testing relevant products as well as developing detailed specifications. In addition, the results of the study provide theoretical guidance for determining the causative factors of failure modes in microelectromechanical systems.

## Figures and Tables

**Figure 1 micromachines-15-00574-f001:**
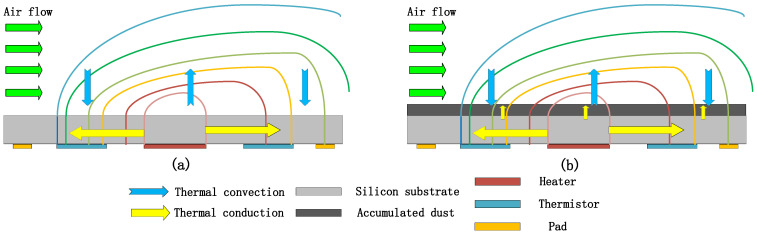
(**a**) Uncontaminated ideal state. (**b**) Contaminated state.

**Figure 2 micromachines-15-00574-f002:**
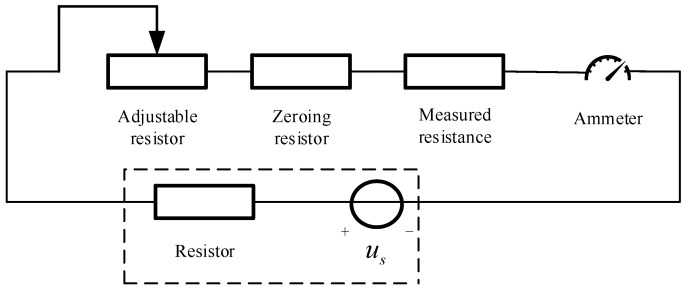
Internal schematic diagram of the multimeter.

**Figure 3 micromachines-15-00574-f003:**
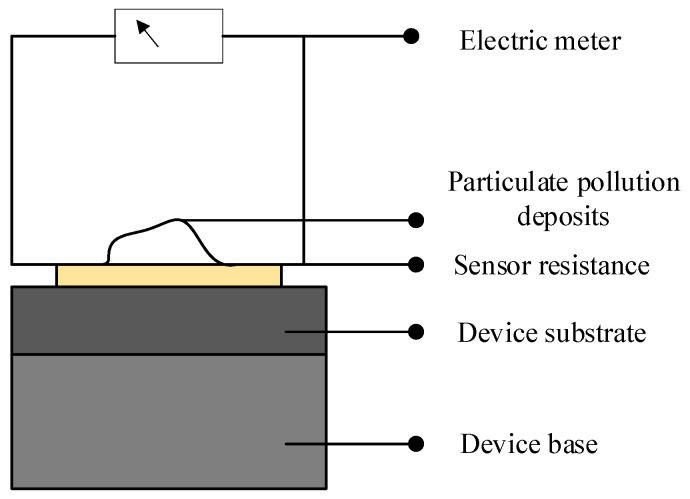
Schematic diagram of sand and dust pollution.

**Figure 4 micromachines-15-00574-f004:**
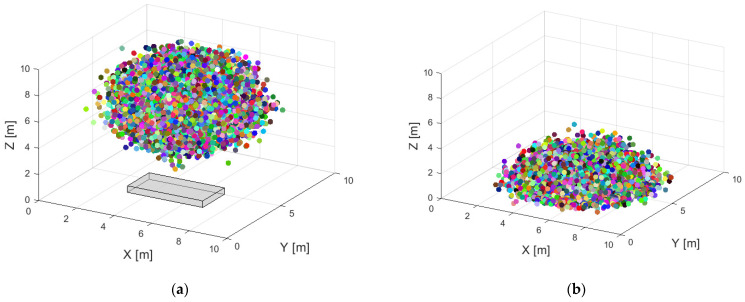
Particle deposition model. (**a**) Dustfall begins; (**b**) end of dust reduction.

**Figure 5 micromachines-15-00574-f005:**
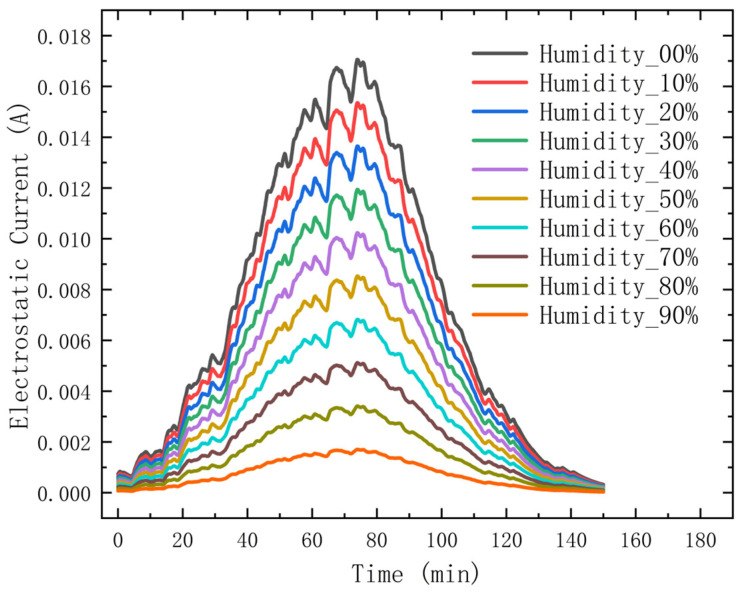
The relationship between electrostatic effect and relative humidity.

**Figure 6 micromachines-15-00574-f006:**
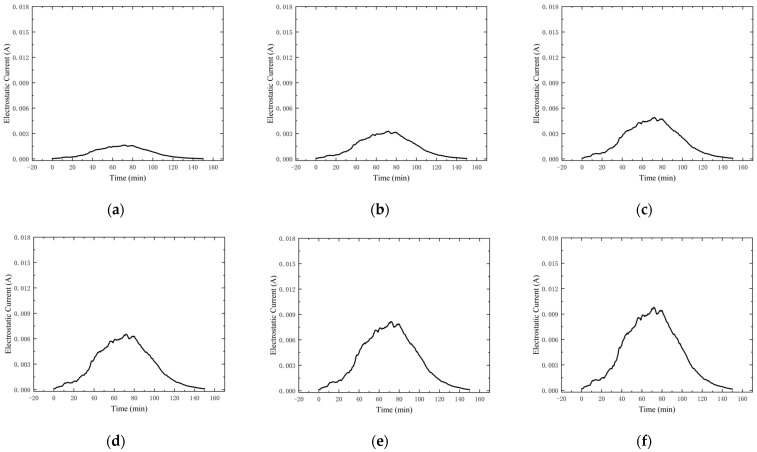
Relationship between electrostatic effect and particle size. (**a**) Particle size: 10 µm. (**b**) Particle size: 20 µm. (**c**) Particle size: 30 µm. (**d**) Particle size: 40 µm. (**e**) Particle size: 50 µm. (**f**) Particle size: 60 µm. (**g**) Particle size: 70 µm. (**h**) Particle size: 80 µm. (**i**) Particle size: 90 µm.

**Figure 7 micromachines-15-00574-f007:**
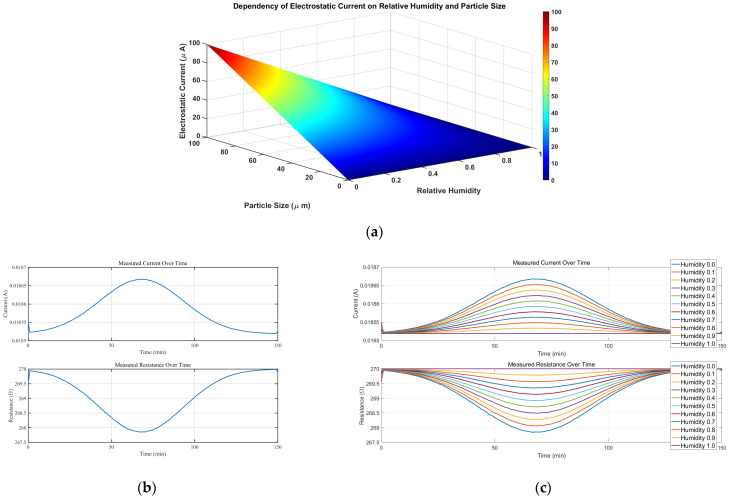
(**a**) The relationship between the electrostatic effect and relative humidity and particle size. (**b**) Single-variable resistance and current measurement. (**c**) Multi-variable resistance and current measurement.

**Figure 8 micromachines-15-00574-f008:**
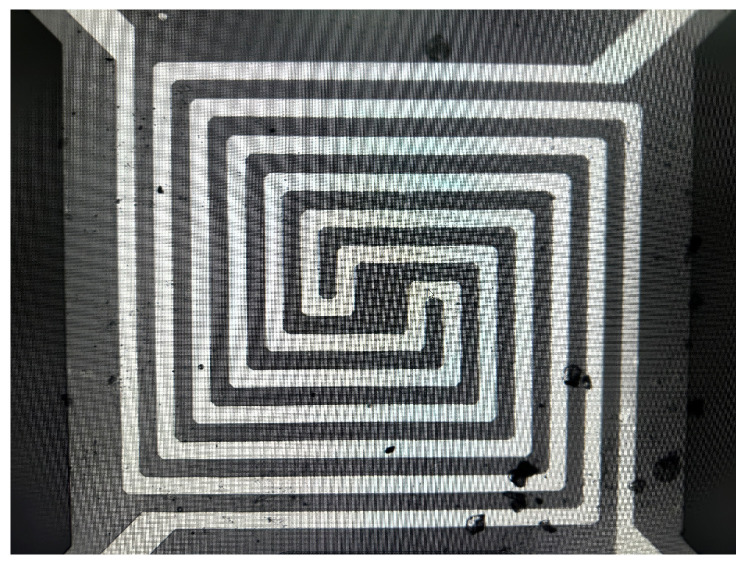
MEMS sensitive structure with a four-ended solid support beam structure.

**Figure 9 micromachines-15-00574-f009:**
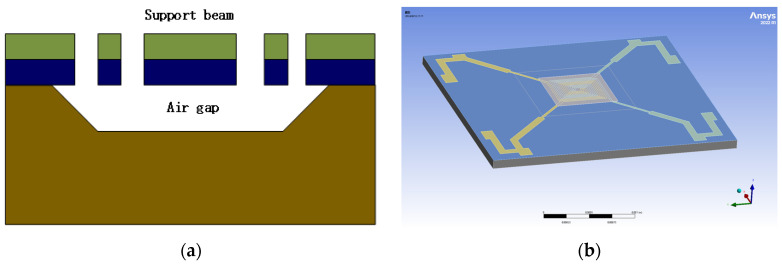
(**a**) Structural section of four-end fixed-support beam. (**b**) Thermistor 3D model drawing.

**Figure 10 micromachines-15-00574-f010:**
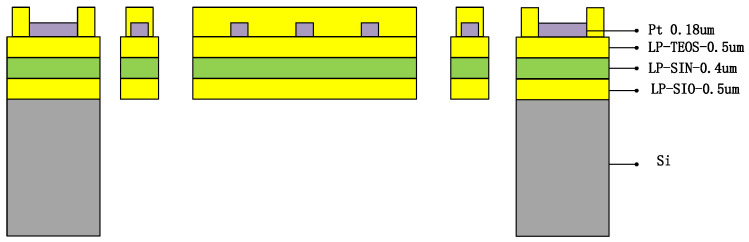
MEMS thermistor cross-section.

**Figure 11 micromachines-15-00574-f011:**
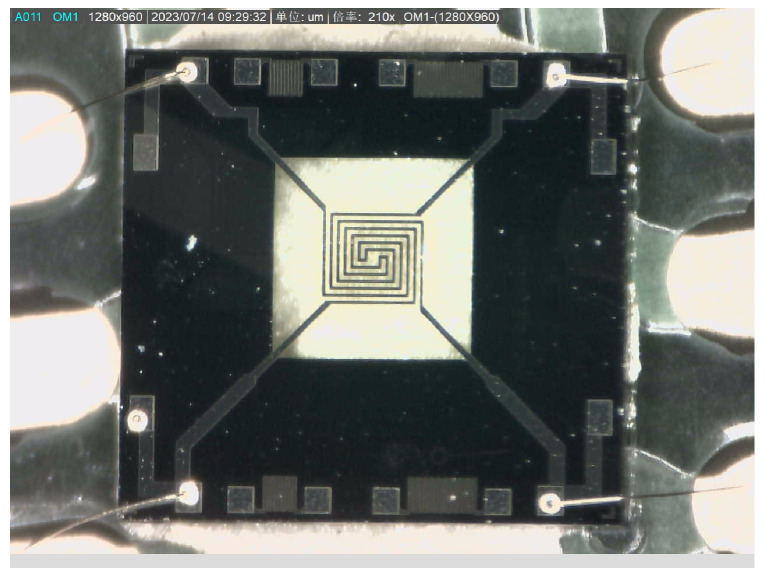
MEMS thermistor physical diagram.

**Figure 12 micromachines-15-00574-f012:**
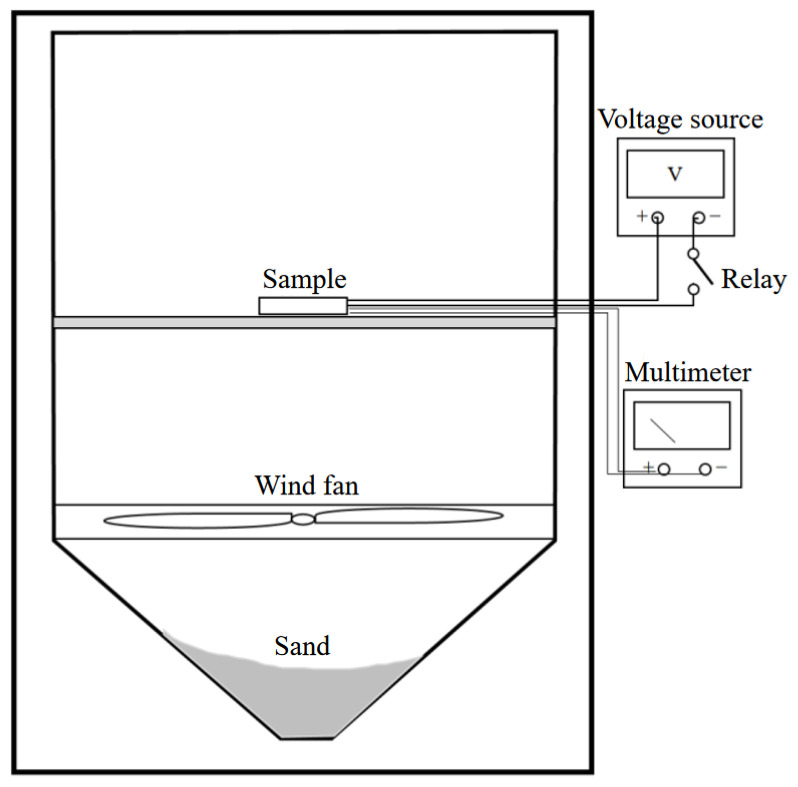
Sand and dust pollution test platform.

**Figure 13 micromachines-15-00574-f013:**
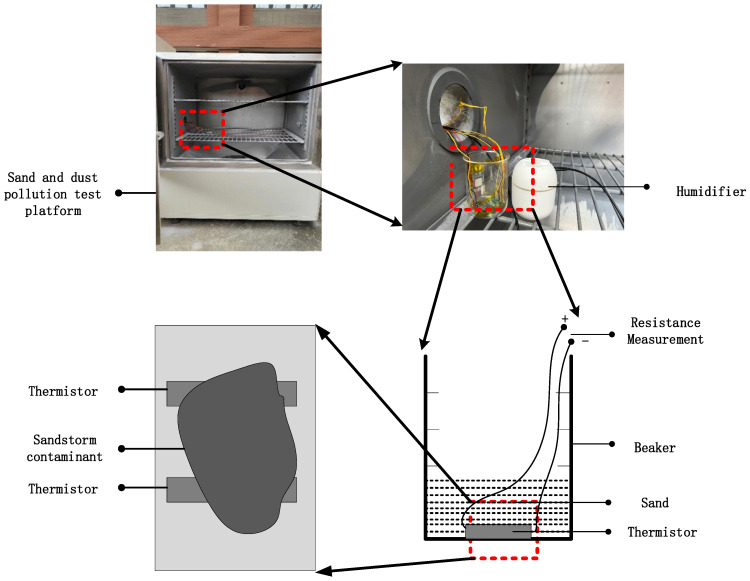
Test environment.

**Figure 14 micromachines-15-00574-f014:**
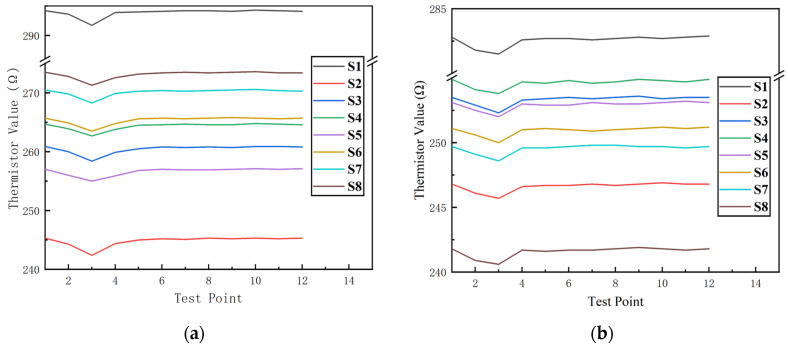
(**a**) Unhumidified sample group. (**b**) Humidification sample set.

**Figure 15 micromachines-15-00574-f015:**
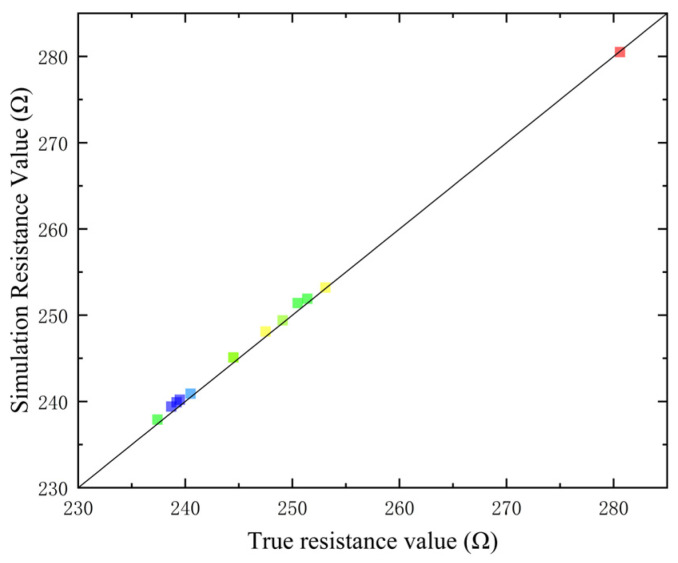
Data comparison.

## Data Availability

The data that support the findings of this study are available from the corresponding authors upon reasonable request.

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
