# Peer review of "Characterization of Sand and Dust Pollution Degradation Based on Sensitive Structure of Microelectromechanical System Flow Sensor†"

_micromachines, 2024, doi:10.3390/mi15050574_

Round 1
Reviewer 1 Report
Comments and Suggestions for Authors
Author(s) provided a method to analyze the accuracy failure of MEMS flow sensors under sand and dust pollution. Author(s) also proposed a combined model for numerical simulation and following test experiments. The model looks interesting, and might be useful for MEMS failure diagnose.
1. In high dust pollution environment, the MEMS sensor should be able to dedusting depend on electrostatic repulsion or its function surface. We might not consider dust pollution problem.
2. For numerical simulation, author(s) employed MATLAB software. We know that MATLAB contains lots of models and mathematical formulas for most of the applications. Did your established a new model using equations (1) to (7) without using existing MATLAB functions/ formulas?
3. The color for accumulated dust and heater look nearly the same (Fig. 1), which makes readers misunderstanding.
4. Ranging from 10-90%, the charge curves are the same except for the amplitude. For most of the cases, the curve obtained by numerical simulation will be like a normal distribution curve without fluctuation.
5. For pollution test, the previous simulation results were not directly verified by the experiments. It is better to obtain similar curves in Fig. 5 and Fig. 6.
6. Author(s) established a test platform (Fig. 12). How did you ensure the dust and humidity were the same in all locations of the chamber? The dust was blown up simply by the wind fan, so how did author could ensure the “pollution level” kept to be a constant level among tests?
Comments on the Quality of English LanguageMinor editing of English language required
Author Response
请参阅附件

Reviewer 2 Report
Comments and Suggestions for Authors
Dear Authors,
Thank you for submitting your manuscript. Your work is well-conceived and the experiments conducted are commendable. However, there are some areas that require modifications. After careful review, I would like to offer several suggestions for revision:
This version keeps the appreciation for the submission and the work done while clearly leading into the feedback section.
1. Graph Consolidation on Humidity Effects: Please consolidate all graphs related to the effect of humidity into one comprehensive figure. This figure should clearly illustrate the range of humidity values from 10% to 90%, providing a unified view of the data.
2. Clarification on Current Measurement and Units: It appears there is a discrepancy in the units presented on the Y-axis of your graphs. The term 'charge' is used, yet the unit is listed as 'Ampere' (A). Please verify and correct the units to accurately reflect the measured quantity.
3. Equation 5 - Threshold Time: Could you provide a more detailed explanation of how the threshold time is derived in Equation 5? A step-by-step elucidation would greatly enhance understanding.
4. Particle Size and Real-Life Scenarios: How does the particle size of the dust used in your experiments compare with those encountered in real-life scenarios? Please cite relevant sources to support your comparison. Additionally, is there a possibility that these particles loosely attach to the sensor surface? Could the electrostatic interactions vary based on the nature of contact with the dust?
5. Figure 7 - Z-Axis Annotation: As previously mentioned, there seems to be a discrepancy in the annotation for the Z-axis, where 'charge' is indicated, yet the unit is 'Ampere'. Please rectify this inconsistency.
6. Usage of Micron Symbol: Throughout the manuscript, please replace "um" with the correct micron symbol (µm) for consistency and accuracy.
7. Figure 8 - Annotations and Adhesion Layer: Could you enhance the annotations in Figure 8 to better align with the manuscript text? Regarding the Pt layer of 180 nm, it would be beneficial to specify if Ti was used as the adhesion layer. Do you have a Scanning Electron Microscope (SEM) image of the Pt microstructure to support your findings?
8. Figure 9 - 3D Cross-Section Schematic: A 3D cross-section schematic of the structure discussed in Figure 9 would greatly aid in understanding. Please consider including this in your revision.
9. Inclusion of Specific Data in Conclusion: To strengthen your conclusion, please include specific numbers, such as particle sizes and resistance thresholds. This would provide a clearer standard for testing and make your findings more relatable and applicable.
Best regards,
Author Response
请参阅附件

Round 2
Reviewer 1 Report
Comments and Suggestions for Authors
Author(s) addressed all my questions, so this manuscript, I advised, could be accepted in this version.
Reviewer 2 Report
Comments and Suggestions for Authors
There is a small typo at line 413. Change to the, he as of now.
The rest of the manuscript looks good. Thanks for updating the manuscript.